# Design and Experiment of Horizontal Transplanter for Sweet Potato Seedlings

**Wei Yan [1,2], Mingjuan Hu [1], Kun Li [1], Jia Wang [1] and Wenyi Zhang [1,*]**

[1] Nanjing Institute of Agricultural Mechanization, Ministry of Agriculture and Rural Affairs, Nanjing 210014, China; yanwei@caas.cn (W.Y.); huminjuan@caas.cn (M.H.); likun03@caas.cn (K.L.); 82101196212@caas.cn (J.W.)

[2] Graduate School, Chinese Academy of Agricultural Sciences, Beijing 100083, China

\* Correspondence: zhangwenyi@caas.cn

**Abstract:** In view of the problems of the difficulties faced in horizontal planting; high labor intensity, poor operation quality and low economy in the existing sweet potato bare seedling transplanting technology and equipment, combined with the agronomic requirements of sweet potato planting and transplanting, a sweet potato bare seedling horizontal transplanter is designed, which can realize multiple processes such as rotary tillage and ridging, ditching and transplanting, covering soil and standing seedlings. The main parameters affecting the working performance and operation effect of the whole machine are analyzed, determining the relevant position and motion parameters, taking the forward speed of machines and tools, spacing of the ribbons and spiral speed as the influencing factors of the performance test, selecting the qualified rate of plant spacing as the evaluation index, and designing a three-factor and three-level orthogonal test. The test results show that the primary and secondary order of the significance of the qualified rate $Z$ of plant spacing is $B$, $A$ and $C$. The better horizontal combination of influencing factors is that the forward speed of machines and tools is $0.17 \text{ m·s}^{-1}$, the spacing of the ribbons is 110 mm, and the spiral speed is 170 rpm. The field test results showed that the average $Z$-value of plant spacing qualification rate under the optimal factor level combination was 91.87%, which met the relevant technical standards and agronomic requirements.

**Keywords:** sweet potato; bare seedling; horizontal transplanter

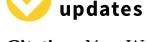



## 1. Introduction

China is the world's largest producer of sweet potato, with a perennial planting area of about $5.33 \times 10^8$ ha, accounting for about 60% of the world production, with an annual production of $8.5\% \times 10^7$ t, accounting for about 79% of the world production [1]. Sweet potato is an important cash crop and has good health care function. With the improvement of people's living standards and the enhancement of health care awareness, the proportion of fresh sweet potatoes is increasing year by year, which puts forward higher requirements for the commodity of sweet potato production and consumption. In recent years, with the rapid development of the social economy and the transfer of rural labor, the labor cost has increased. The demand for mechanized transplanting technology and equipment in sweet potato producing areas is increasing, and it is urgent to adopt mechanical replacement rather than relying on a large number of labor [2]. Due to the growth characteristics of sweet potato, it is transplanted by bare seedlings. The labor volume of sweet potato transplanting accounts for about 23% of the whole production process. At present, the mechanized transplanting area of sweet potatoes is less than 3%. A large number of sweet potato transplanting still rely on heavy manual labor. The manual operation cost is high, and the operation quality is difficult to be standardized and unified, resulting in poor commercialization, low planting efficiency, and disconnection from the market consumption demand, which seriously restricts the high-quality and healthy development of China's sweet potato industry. As the transplanting of sweet potato requires bare seedling

transplanting, it cannot be transplanted in cavity trays like vegetables and other crops. The bare seedling of sweet potato has complex plant morphology, staggered leaves and vines, and poor morphological consistency; thus, it is difficult to transplant mechanically. At present, manual transplanting is still the main operation. Compared with other operations such as ridging, vine cutting, and harvesting, transplanting has become a shortcoming in the sweet potato production process.

At present, most sweet potato transplanting equipment in China is restructured from vegetable, tobacco, sugar beet, and other transplanting machinery, and the main planting mechanisms are chain clip type and duckbill type. The chain clip type can realize oblique planting, but it is easy to leak seedlings, and the artificial seedling feeding is labor-intensive, and the planting quality is average. The duckbill type can realize oblique planting, but the seedling guide tube parts are easy to entangle the seedlings, with high requirements for seedlings, and planting quality is poor. At present, only Japan's Iseki Co., Ltd., (Matsuyama, Japan) produces special-purpose machinery that is matched with sweet potato transplanting, which adopts the way of artificial feeding and clamping planting, but the planting efficiency is low, the soil adaptability needs to be improved, and it is not suitable for large-scale planting. There are few other sweet potatoes bare seedling planting techniques that can be used for reference abroad, and its production mode and soil properties are different from those in China, which makes it difficult to adapt to the actual needs of sweet potato production. Therefore, to solve the problem of sweet potato mechanized planting in China, we can only rely on independent innovation and thought breakthroughs [3].

Aiming at the mechanization of sweet potato transplanting, this paper developed a horizontal transplanting machine for bare seedlings of sweet potato, which adopts key technologies such as positioning and fixed-spacing seedlings, flexible clamping, full-process control, ditching and fixed depth, low-position seedlings, cushioning and exposing tops, spiral covering soil, and suppressing seedlings. It can effectively reduce labor intensity and improve operation efficiency and planting quality [4–6]. Further, this study will help to determine the optimal combination of factors affecting the operation quality of sweet potato bare seedling horizontal transplanter.

## 2. Materials and Methods

### 2.1. Planting Agronomic Requirements

Sweet potato planting methods mainly include 5 methods: horizontal planting; oblique planting; boat bottom planting; straight planting; and pressing rattan planting, as shown in Figure 1 [7].

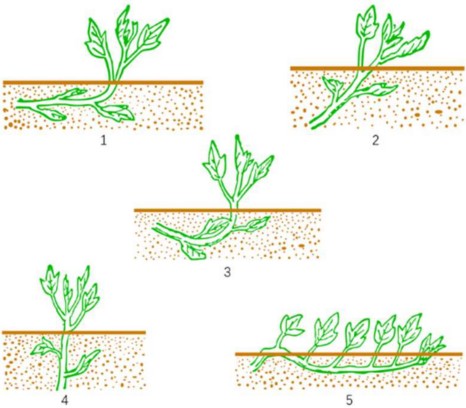

**Figure 1.** Schematic diagram of sweet potato planting method: 1. Horizontal method; 2. Oblique method; 3. Boat bottom planting; 4. Straight planting; 5. Pressing rattan planting.

Compared with the other 4 planting methods, horizontal planting has the advantages of fewer empty nodes, higher yield, more and uniform tubers, and was the dominant practice in major areas. The length of the bare seedlings planted horizontally was 20–30 cm,

the length of the planting soil was 10–15 cm, the planting depth was about 8–10 cm and the part of the planting soil should be in a horizontal posture [8,9], as shown in Figure 2. In the figure, $l_1$ is the planting distance, which is generally 150–300 mm according to different needs; $l_2$ is the horizontal length of the planting soil, which requires 100 mm (1 ± 10%); $h_1$ is the depth of the planting soil, which is required to be 100 mm (1 ± 10%).

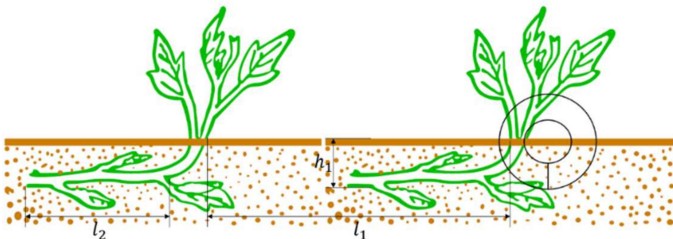

**Figure 2.** Schematic diagram of horizontal planting of bare sweet potato seedlings.

Due to the complex shape and poor consistency of bare sweet potato seedlings, it was difficult to control the posture of bare sweet potato seedlings during the planting process. In addition, horizontal planting has unique planting requirements such as long planting soil length, deep planting depth, and the need to expose the top 3 leaves; thus, it is difficult to solve the problem mechanically. At present, they mainly rely on manual planting operations.

### 2.2. Design of Transplanter

The schematic diagram of the overall structure of the sweet potato bare seedling horizontal transplanter is shown in Figure 3. It is mainly composed of the tractor, rotary tiller, ridge body forming plate, seat, transmission box, upper seedling protection belt, lower seedling conveying belt, screw mechanism, suppression mechanism, seedling tray, support mechanism, etc. When working, the implement is rigidly connected to the tractor through a 3-point hitch.

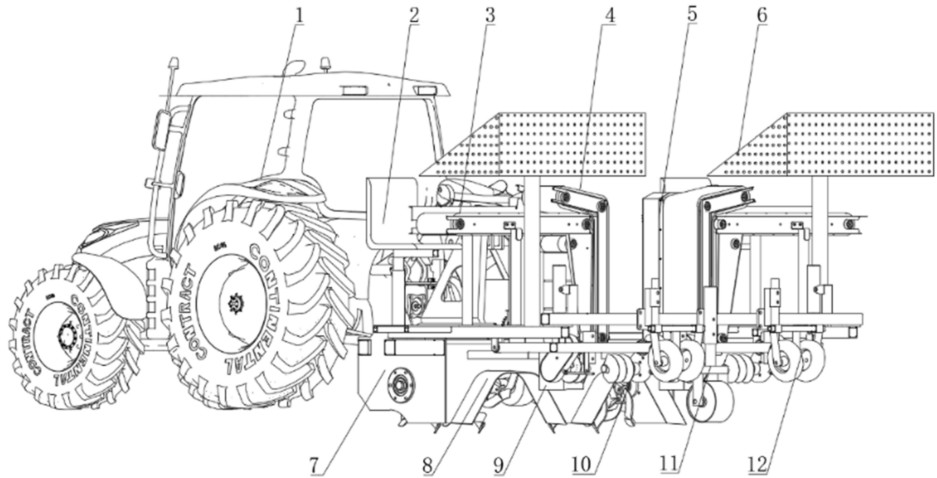

**Figure 3.** Schematic diagram of horizontal transplanter for bare sweet potato seedlings: 1. Tractor; 2. Seat; 3. Lower seedling transport belt; 4. Upper seedling protection belt; 5. Transmission box; 6. Seedling tray; 7. Rotary tiller; 8. Ridge body forming plate; 9. Ditch opener; 10. Screw mechanism; 11. Support Wheel; 12. Suppression Wheel.

### 2.3. Working Principle

The transmission system of the implement is shown in Figure 4. The traction frame was connected to the tractor by means of a 3-point hitch. The rear power output shaft of the tractor was connected to the gearbox through a universal joint to transmit the power

to the rotary tiller and the reducer. The reducer transmits the power to the reversing box and the transmission box, the transmission box transmits the power to the upper seedling protection belt and the lower seedling conveying belt, and the commutation box transmits the power to the screw mulching mechanism to complete the mulching and planting.

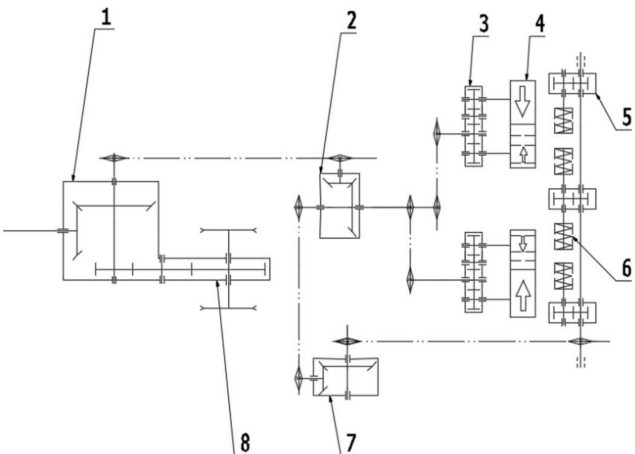

**Figure 4.** Transmission system of sweet potato bare seedling horizontal transplanter: 1. Gearbox; 2. Speed reducer; 3. Conveyor transmission box; 4. Conveyor Belt; 5. Screw mechanism transmission box; 6. Screw mechanism; 7. Reversing box; 8. Rotary tillage system.

When the machine is working, the sweet potato bare seedling horizontal transplanter moves forward with the tractor, and the rotary tiller cooperates with the ridge body forming plate to complete the ridge operation. At the same time, the ditch opener opens the ditch on the ridge body, and the potato seedlings are manually placed in the groove on the lower seedling conveying belt, and the potato seedlings are conveyed horizontally. When the seedlings are transported from the horizontal to the vertical direction, the lower seedling belt and the upper seedling guard belt will hold the seedlings and transport them synchronously. When conveyed to the bottom of the seedling conveying belt, the lower seedling conveying belt is separated from the upper seedling protection belt, and the potato seedlings are freely dropped from the groove of the lower seedling conveying belt to the opened seedling ditch under the action of inertial force. During the falling process of the potato seedlings, the spiral soil covering mechanism pushes the soil on both sides of the seedling ditch into the seedling ditch to complete the operations of covering the tops with soil and covering the seedlings. At the same time, the pressing wheel is used to suppress and solidify the seedlings to complete the horizontal planting of bare sweet potato seedlings.

*2.4. Main Technical Parameters*

The main technical parameters of sweet potato bare seedling horizontal transplanter are shown in Table 1.

**Table 1.** Main technical parameters of transplanter.

| Project | Parameter |
| --- | --- |
| Working width/mm | 2200 |
| Supporting power/kW | ≥66.19 |
| Number of blades | 52 |
| Tillage depth/mm | 180 |
| Turning radius/mm | 240 |
| Operating speed/m·s$^{-1}$ | ≥0.1 |
| Number of job lines | 2 |

### 2.5. Design of Ridge Forming Part

The ridge body forming part is mainly composed of a ridge body forming plate and a connecting plate, as shown in Figure 5. In order to ensure the possibility of the soil and the compactness of the ridge body, the ridge body forming plate was designed with a "flare mouth". The dimensions of the ridge body forming plate at the entrance were ridge height 350 mm, ridge bottom width 800 mm, ridge top width 350 mm. The size of the ridge body forming plate at the outlet was 300 mm in height, 700 mm in width at the bottom of the ridge, and 300 mm in width at the top of the ridge.

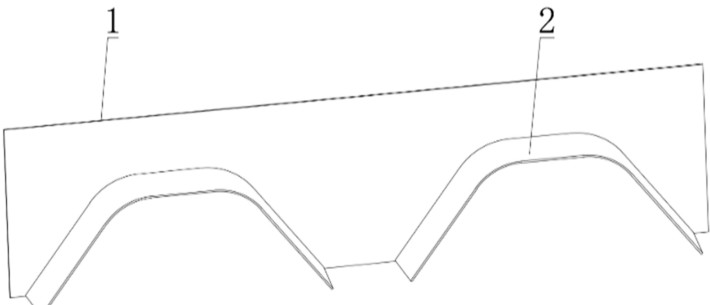

**Figure 5.** Ridge body forming parts: 1. Connecting plate; 2. Ridge body forming plate.

### 2.6. Design of Belt Seedling Conveying Part

The belt-type seedling conveying components mainly include the upper seedling protection belt, the transmission box, the lower seedling conveying belt, the seedling swinging trough, the connecting plate, the transmission roller, etc., as shown in Figure 6. The upper seedling protection belt and the lower conveyor belt are canvas sponge belts, the thickness of the upper seedling protection belt is 10 mm, and the thickness of the lower conveyor belt is 15 mm. If the width of the conveyor belt was smaller than the length of the seedlings, it would lead to unstable seedling transport and affect the planting effect; thus, the width of the conveyor belt was designed to be 250 mm according to the length of the transplanted seedlings. There is a groove for placing seedlings on the lower conveyor belt, and the seedlings are placed in the groove manually. The shape of the end surface of the groove is a semicircle. According to the experimental data on sweet potato stem thickness and sweet potato seedling width in the previous stage, the diameter of the groove was set to 10 mm, and the spacing between adjacent grooves was 200 mm; the transmission box drives the upper seedling protection belt and the lower conveyor belt to rotate synchronously through the transmission roller. In order to avoid the phenomenon of pushing the seedlings when the lower conveyor belt and the upper seedling protection belt clamp the sweet potato seedlings, the upper seedling protection belt is designed to be inclined, and the angle between the inclined section and the vertical section was 105°. According to the design strength of the sponge rotation, the diameter of the designed transmission roller was 87 mm. In order to reduce the influence of the height of the sweet potato seedlings falling freely to the bottom of the trench on the planting quality, the distance from the release position of the sweet potato seedlings to the top of the trench opener was designed to be 40 mm.

Ideally, after the machine is running stably, the number of sweet potato seedlings falling into the seedling ditch at the same time is equal to the number output by the seedling conveying components.

$$\frac{v_1 \Delta t}{l_1} = \frac{v_2 \Delta t}{l_2} \tag{1}$$

is

$$\frac{v_1}{l_1} = \frac{v_2}{l_2} \tag{2}$$

In the formula, $v_1$—forward speed of sweet potato belt transplanter, m·s$^{-1}$; $v_2$—linear speed of conveyor belt, m·s$^{-1}$; $l_1$—theoretical planting distance, mm; $l_2$—belt groove spacing, mm; $\Delta t$—unit time, s.

According to agronomic requirements, the machine sets the plant spacing $l_1$ to 250 mm. According to the artificial seedling placement speed, the planting speed $v_1$ of the belt transplanter is generally 0.36~0.9 km·h$^{-1}$ (0.1~0.25 m·s$^{-1}$), and the spacing of the belt grooves is set to 200 mm, then the seedling conveying belt speed $v_2$ is $8.8 \times 10^{-2}$~$2.0 \times 10^{-1}$ m·s$^{-1}$.

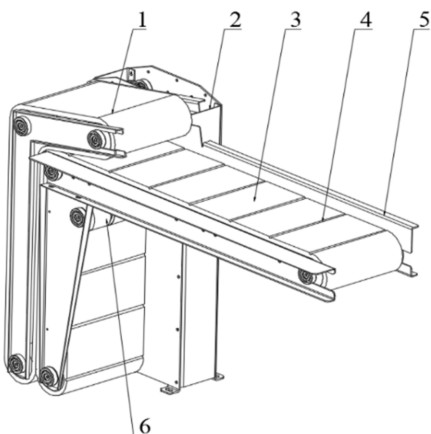

**Figure 6.** Belt Seedling Conveying Parts: 1. Upper seedling protection belt; 2. Transmission box; 3. Lower conveyor belt; 4. Seedling trough; 5. Link plate; 6. Drive roller.

*2.7. Design of Ditching Depth Parts*

The ditching and depth-fixing part is the working part in contact with the soil [10–12]. It is mainly to open the seedling ditch to prepare for the implantation of sweet potato seedlings, mainly including ditching sheets and connecting plates, as shown in Figure 7. The ditching and depth-fixing components were fixed on the frame through the connecting plate. According to the agronomic requirements of sweet potato planting, the height of the designed ditch opener was 130 mm, and the ditching depth was adjustable from 0 to 120 mm. According to the physical characteristics of seedling transportation, in order to minimize the collision between the sweet potato seedlings and the sidewall of the opener when they fall, the trench width was set to 70 mm. During ditching operation, in order to ensure the better soil-breaking performance of the ditching and depth-fixing components and to promote the smooth flow of soil along both sides of the ditching sheet. The angle between the 2 ditching pieces was designed to be 35°, and a "clean area" was formed inside the ditching and depth-determining components, which can effectively reduce the influence of soil, weeds, etc., on the planting depth. At the same time, its position can be adjusted back and forth within the range of 0~50 mm.

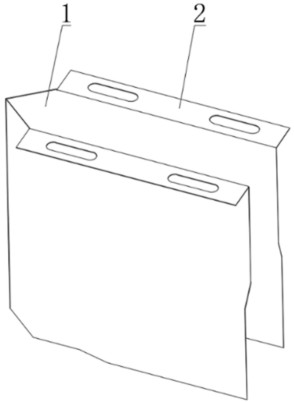

**Figure 7.** Ditching depth parts: 1. Ditching piece; 2. Connecting plate.

### 2.8. Design of Mulching and Planting Parts

The pad tip cover is mainly composed of a support frame, a bearing seat, a screw mechanism, a drive shaft, a screw rod, and a chain shell, as shown in Figure 8. The drive shaft drives the screw mechanism through the chain drive to complete the mulching and planting operation. By adjusting the support length of the screw, the relative position of the centerline of the screw mechanism and the outer edge of the conveyor belt (spacing of the ribbons) is adjusted.

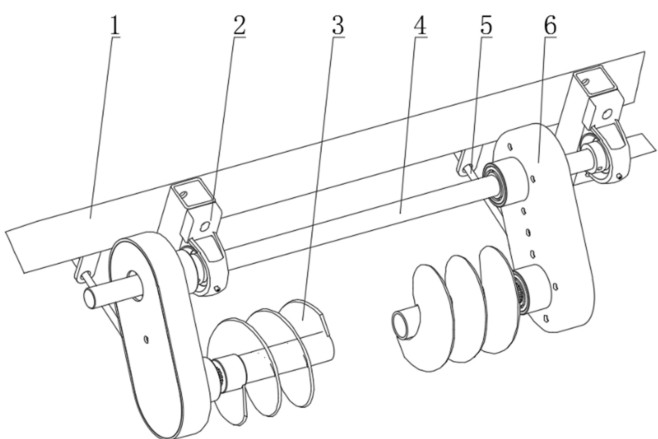

**Figure 8.** Pad tip cover seedling parts: 1. Support frame; 2. Bearing seat; 3. Screw; 4. Transmission shaft; 5. Screw rod; 6. Chain cover.

The soil covering of the screw mechanism needs to make the soil particles move outward in the axial direction, and the direction of the movement speed of the particles is from the inside to the outside (with the normal deflection of the helical surface by a friction angle), and the condition is:

$$\tan \alpha \leq 1 / \tan \varphi \tag{3}$$

where $\alpha$ is the helix angle (°); $\varphi$ is the friction angle between the soil and the steel plate (°). Since the helix angle of each point on the blade is unequal, the helix angle at the smallest radius is the largest; thus, as long as the helix angle at the inner diameter satisfies the above conditions [13].

The theoretical mud transport volume of the screw mechanism is the volume of the seedling ditch opened during the ditching operation per unit time. The opened seedling ditch can be regarded as a cuboid, and the mud transportation required to fill the seedling ditch can be expressed as:

$$V = bvh \tag{4}$$

In the formula: $b$ is the working width (m); $v$ is the forward speed (m·s$^{-1}$); $h$ is the tillage depth (m).

According to the agronomic requirements of the sweet potato transplanter, the operating speed of the tractor is generally in the range of 0.1 to 0.25 m·s$^{-1}$, the ditching width of the opener is 70 mm, and the ditching depth is 80 mm. The amount of mud transported is about $5.6 \times 10^{-4} \sim 1.4 \times 10^{-3}$ m$^3$·s$^{-1}$.

Referring to the design method of the open screw conveyor, in order to ensure that the spiral mud is transported smoothly and does not appear to be clogged, the design of the spiral soil cover should meet the following requirements [14,15]:

(1) The screw conveying capacity must be greater than the amount of soil feeding, otherwise it will cause sludge;

(2) The soil cannot produce jumping and rolling perpendicular to the conveying direction;

(3) The soil is transported along the axial direction, and the axial force and axial velocity are greater than 0.

Referring to the screw conveying mechanism and theoretical design method, the design formula of screw parameters is as follows:

Spiral outer diameter

$$D \geq K \sqrt[2.5]{\frac{V}{\varphi \varepsilon \rho}} \tag{5}$$

Shaft diameter

$$d = (0.2 \sim 0.35)D \tag{6}$$

Spiral speed

$$n \leq n_{max} = \frac{A}{\sqrt{D}} \tag{7}$$

Screw pitch

$$S_{max} = (0.5 \sim 2.2)D \tag{8}$$

In the formula, $V$ is the amount of mud transported, $m^3 \cdot s^{-1}$; $K$ is the comprehensive characteristic coefficient of the mud; $\varphi$ is the filling coefficient; $\varepsilon$ is the slope coefficient; $\rho$ is the bulk density of the conveyed material, $t \cdot m^{-3}$; $A$ is the comprehensive characteristic coefficient.

Due to the symmetrical arrangement of the left and right covering soil spirals, the amount of mud transported by the spiral should be half of that required to fill the seedling ditch. The value of $V$ is $2.8 \times 10^{-4} \sim 0.7 \times 10^{-3}$ $m^3 \cdot s^{-1}$; the topsoil of sweet potato transplanting soil has a certain fluidity, referring to the design standard of the screw conveyor, the estimated soil comprehensive characteristic coefficient $K$ is 0.0415; the filling factor $\varphi$ is taken as 0.4; since the spiral is placed horizontally, the spiral inclination coefficient $\varepsilon$ is taken as 1.0; the comprehensive characteristic coefficient $A$ of the soil is taken as 75; the bulk density $\rho$ of soil is taken as 1.8 $t \cdot m^{-3}$.

From Formula (5), it can be calculated that the outer diameter of the screw $D \geq 172$ mm, considering the quality of the whole machine and the reasonable matching of the tractor, take $D = 180$ mm. From Formula (6), it can be calculated that the diameter of the screw shaft can be between 36 and 63 mm. In order to minimize the entanglement of grass on the shaft and improve the soil-moving capacity, the diameter of the screw shaft is d = 60 mm. According to Formula (8), the range of pitch is 90~396 mm. In order to ensure the smooth conveyance of soil in the screw and prevent mud accumulation, the amount of mud transported by each screw pitch of the screw cover should be greater than or include the sum of the mud input amount of this screw pitch and the previous screw pitch, otherwise, it will cause mud accumulation or lack of mud [16]. Therefore, the variable pitch equal-diameter helix was used in this study, and the pitch increased along the 2 ends of the left and right soil-covering helix towards the sweet potato seedling ditch, and the pitch near the seedling ditch was the largest. According to the structure and configuration size of the whole machine, it was preliminarily determined that the maximum pitch of the first stage was 110 mm, the pitch of the second stage was 100 mm and the pitch of the third stage was 90 mm. Calculated from Formula (7), the maximum working speed was 176 rpm.

## 3. Results

In order to realize the efficient planting of sweet potatoes, the operation effect of the horizontal transplanting machine for bare seedlings of sweet potato was evaluated. According to the design parameters, the performance test of the sweet potato belt type transplanting was carried out [17].

### 3.1. Test Conditions and Equipment

In May 2021, a sweet potato transplanting test was carried out at the sweet potato planting base in Wugang City, Henan Province. The test site was an idle winter field, the soil type was sandy soil, and the soil moisture content was about 15% to 19% (0 to 100 mm). The length of sweet potato seedlings was 250 mm, and the diameter of sweet

potato seedlings was 4~6 mm. The test equipment was mainly composed of Dongfanghong 904 tractor, sweet potato bare seedling horizontal transplanter, moisture meter, calculator, ruler, tape measure, tachometer, etc.

### 3.2. Experimental Design and Methods

The main parameters that affect the working performance and operation effect of the whole machine were selected: the forward speed of the machine $v$, the screw speed $n$ and spacing of the ribbons $\tau$ as the test factors, and the plant distance $Z$ as the evaluation index to characterize the operation quality of the machine. According to the above experimental plan, a single-factor experiment was carried out. The factors affecting the qualified rate of planting distance and the range of values were determined, and a three-factor and three-level orthogonal experiment was designed ($L_9(3^4)$) [18,19]. The factor levels are shown in Table 2.

**Table 2.** Factors and levels of orthogonal test.

| Test Level | The Forward Speed of the Machine $v$/m·s$^{-1}$ | Spacing of the Ribbons $\tau$/mm | The Screw Speed $n$/rpm |
|---|---|---|---|
| 1 | 0.15 | 110 | 130 |
| 2 | 0.17 | 120 | 150 |
| 3 | 0.19 | 130 | 170 |

In the experiment, we calculated the planting spacing qualification rate $Z$ under the conditions of each parameter combination by adjusting the different forward speed of the machine, spacing of the ribbons and screw speeds. The operational performance of the machine was evaluated by analyzing the planting spacing qualification rate $Z$. Each test group was repeated three times to take the average value. The test evaluation index was based on GB/T 5262 [20]. A total of three measuring areas were randomly selected in the experimental site; each measuring area must contain two adjacent working widths, select a row for each measuring area, and measure 120 plant spacing continuously. The measured plant spacing was within D ($1 \pm 10\%$), which was qualified, and the percentage of qualified plant spacing to the total number of measurements was the qualified plant spacing rate, which was calculated according to Formulas (9) and (10) [21–23].

$$Z_i = \frac{N_{zi}}{N} \times 100 \tag{9}$$

$$Z = \frac{\sum_{i=1}^{n} z_i}{n} \times 100 \tag{10}$$

In the formula:
$N_{zi}$—the qualified number of plant spacing in the testing area;
$N$—the total number of samples measured in the detection area, $N = 120$;
$Z_i$—the qualified rate of plant spacing in the detection area, the unit is percentage (%);
$Z$—the qualified rate of plant spacing, the unit is percentage (%);
$n$—the number of detection areas, $n = 3$.

### 3.3. Test Results and Analysis

The experimental results measured according to the above orthogonal performance test scheme are shown in Table 3, where *A* (The forward speed of the machine), *B* (Spacing of the ribbons), and *C* (The screw speed) were the level values of $v$, $\tau$, and $n$, respectively.

**Table 3.** Results and orthogonal test.

| Test Number | | Test Factor | | | Test Index |
|---|---|---|---|---|---|
| | | A | B | C | Plant Spacing Qualification Rate Z/% |
| 1 | | 1 | 1 | 1 | 91.56 |
| 2 | | 1 | 2 | 2 | 87.63 |
| 3 | | 1 | 3 | 3 | 88.47 |
| 4 | | 2 | 1 | 2 | 93.46 |
| 5 | | 2 | 2 | 3 | 94.28 |
| 6 | | 2 | 3 | 1 | 90.26 |
| 7 | | 3 | 1 | 3 | 93.42 |
| 8 | | 3 | 2 | 1 | 89.74 |
| 9 | | 3 | 3 | 2 | 86.59 |
| z | $k_1$ | 89.22 | 92.81 | 90.52 | |
| | $k_2$ | 92.67 | 90.55 | 89.06 | |
| | $k_3$ | 89.92 | 88.44 | 92.06 | |
| | $k_4$ | 3.45 | 4.37 | 2.82 | |

According to the numerical analysis of the range $R$ of each factor in Table 3, it can be seen that for the evaluation index $Z$, the order of significance of the influence of each factor was $B$, $A$, $C$ and the optimal level combination of influencing factors was $A_2B_1C_3$.

SPSS 21.0 data processing software was used to conduct variance analysis on the test results [24–26]. The variance analysis results are shown in Table 4. It can be seen from Table 4 that the sum of squares of the error term was much smaller than the sum of squares of the influencing factors, indicating that the interaction between the experimental factors has no significant influence on the experimental assessment indicators. When analyzing the qualification rate $Z$ of the assessment index plant spacing, it can be seen from the $p$-value that the experimental factors $A$, $B$ and $C$ have a significant impact on the experimental index $Z$; from $F_B > F_A > F_C$, it can be seen that the test factor $B$ had the greatest influence on the test index, the test factor $A$ had the second influence, and the test factor $C$ had the least influence, which was consistent with the range analysis results.

**Table 4.** Analysis of variance.

| Evaluation Indicators | Source of Variation | Sum of Square | Degrees of Freedom | Sum of Mean Squares | F | p |
|---|---|---|---|---|---|---|
| | calibration model | 62.23 | 6 | 10.37 | 64.24 | 0.015 * |
| | A | 20.69 | 2 | 10.35 | 64.06 | 0.015 * |
| Z | B | 28.75 | 2 | 14.37 | 89.00 | 0.011 * |
| | C | 12.80 | 2 | 6.40 | 39.64 | 0.025 * |
| | error | 0.32 | 2 | 0.16 | | |

Note: $R^2 = 0.995$, * is significant ($p < 0.05$).

Based on the results of range analysis and variance analysis, it can be seen that the best combination of parameters for transplanting performance in this experiment was $A_2B_1C_3$, the forward speed of the machine was 0.17 m·s$^{-1}$, the spacing of the ribbons was 110 mm, and the screw speed was 170 rpm.

*3.4. Field Trial*

In order to verify the working performance of the sweet potato belt transplanter under the optimal combination of factors in the above orthogonal test, a field test was conducted in the sweet potato planting base of Wugang City, Henan Province, in June 2021, the test effect was shown in Figure 9. Before the test, adjust the working parameters of the whole machine to the best level combination: the forward speed of the machine was 0.17 m·s$^{-1}$,

the spacing of the ribbons was 110 mm, and the screw speed was 170 rpm. The test results are shown in Table 5.

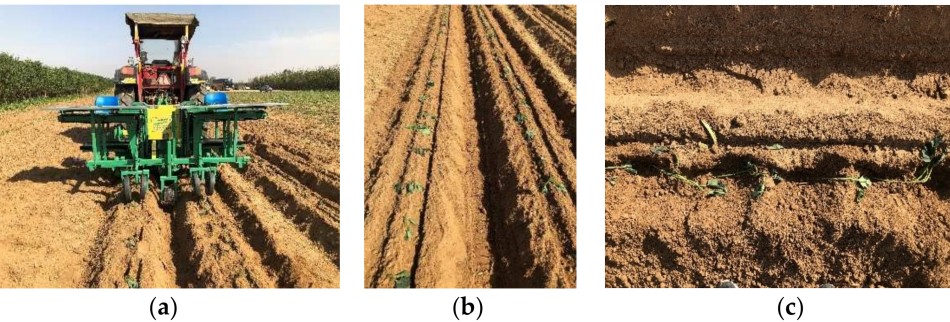

|      (**a**)      |      (**b**)      |      (**c**)      |

**Figure 9.** Field experiment results: (**a**) field experiment; (**b**) experiment result; (**c**) effect of horizontal planting.

**Table 5.** Results of the field test.

| Test Serial Number | Plant Spacing Qualification Rate Z/% |
| --- | --- |
| 1 | 92.36 |
| 2 | 93.62 |
| 3 | 90.56 |
| 4 | 88.67 |
| 5 | 94.28 |
| 6 | 91.75 |
| mean | 91.87 |

The test results in Table 5 show that the average planting distance of the sweet potato bare seedling horizontal transplanter was 91.87%, which is in line with the relevant agricultural machinery industry technology and standards and local agronomic production requirements. The horizontal planting of potato seedlings can be realized, which is beneficial to improve the uniformity and yield of sweet potatoes.

## 4. Discussion

In this paper, the effects of machine advance speed, spacing between spirals, and screw speed on the planting spacing qualification rate were studied. Further optimization and improvement of the machine is needed in later experiments for such indexes as planting depth qualification rate and plant spacing variation coefficient.

During the machine tests, it was found that the tractor occasionally has a wheel slip problem during traction operations, and this problem has an effect on the plant distance. For this problem, future research can use the electric drive control system to solve the error caused by wheel slippage [27].

Compared with the sweet potato transplanter used at this stage, the belt transplanter realizes horizontal planting of seedlings, which improves planting quality, saves labor, and reduces labor intensity. However, seedling feeding is still conducted manually, and the operation efficiency is limited to manual work. Subsequent research on automatic seedling feeding technology is needed to realize automatic seedling feeding and further improve operation efficiency.

## 5. Conclusions

(1) In view of the problems of difficult horizontal planting, high labor intensity, poor operation quality and low economy in the existing sweet potato transplanting technology and equipment, a horizontal transplanting machine for bare sweet potato seedlings is designed, which can realize multiple processes such as rotary tillage and ridge raising,

ditching and transplanting, and soil-covered seedlings, providing a new method for horizontal transplanting of bare sweet potato seedlings;

(2) Analyze the main parameters affecting the working performance and working effect of the whole machine, determine the relevant position and motion parameters, and select the optimal working parameter combination through the performance test. The test results show that the primary and secondary order of the $Z$ significance of the qualified rate of plant spacing for the impact evaluation index is $B$, $A$, $C$ and the optimal level combination of influencing factors is $A_2 B_1 C_3$, the forward speed of the machine is 0.17 m·s$^{-1}$, and the spacing of the ribbons is 110 mm, the screw speed is 170 rpm;

(3) The field test results showed that: under the optimal combination of factor levels, the $Z$-average rate of plant spacing was 91.87%, which met the relevant technical standards and agronomic requirements.

**Author Contributions:** Conceptualization, W.Y. and M.H.; methodology, W.Y.; software, K.L.; validation, W.Y., M.H. and W.Z.; formal analysis, W.Y.; investigation, K.L.; resources, W.Z.; data curation, W.Y.; writing—original draft preparation, W.Y.; writing—review and editing, J.W.; visualization, W.Z.; supervision, J.W.; project administration, W.Z. All authors have read and agreed to the published version of the manuscript.

**Funding:** This research was funded by China Agriculture Research System of MOF and MARA (Grant No. CARS-10-B19) and the Key R & D Plan of Jiangsu Province (Grant No. BE2021311).

**Institutional Review Board Statement:** Not applicable.

**Informed Consent Statement:** Not applicable.

**Data Availability Statement:** The datasets used and/or analyzed during the current study are available from the corresponding author on reasonable request.

**Conflicts of Interest:** The authors declare no conflict of interest.

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
