# Peer review of "Design and Experiment of Horizontal Transplanter for Sweet Potato Seedlings"

_agriculture, doi:10.3390/agriculture12050675_

Round 1
Reviewer 1 Report
- If the power to the conveying belts was transferred with the help of tractor PTO drive rather than any ground wheel, doesn't it affected the performance of the machine due to wheel slip experienced in the fields? How this was taken care of?
- Experimental design: What about the other factors such as moving belt speed, and rotary tiller parameters? Why these factors were not taken into account.
- How the optimum level combination was interpreted from Table 3?
- The function of screw covering mechanism needs to be explained?
- Comparison of the developed planter with any conventionally available potato seedling transplanting machine is missing.
- Table 1: These (Row numbers 3, 4, 5, 6, and 8) are basically the performance parameters of the machine and not the technical specifications. Please delete it from this section and provide the proper detailed technical specifications of your developed machine.
- Nothing has been discussed about the rotary tiller mounted at the most front. What type of blades were used, rotor speed, depth of tillage etc.
- What is GB/T 5262? Is it a test code? Please give proper reference.
- There should be a space between the numerical value and the unit. For example it should be 20-30 cm not 20-30cm. Please correct it for the entire manuscript.
- Lines 132-153 should be moved to section 2.1.
- Lines 157-158: What do you mean by length of planting and how it is a design criteria for deciding the width of conveyor belt? Please discuss here.
- Grooves should be shown in the Figure 6 for better understanding.
- The headings of sections 3.4.1 and 3.4.2 are same. Please rectify.
- Lines 305-315: Write the entire methodology in the correct format and tense. It seems like some instructions have been given to perform the tests.

Author Response
Point 1: If the power to the conveying belts was transferred with the help of tractor PTO drive rather than any ground wheel, doesn't it affected the performance of the machine due to wheel slip experienced in the fields? How this was taken care of?
Response 1: First of all, the problem you raised is indeed a technical problem that we encountered in the actual production operation. Our solution is that before transplanting, we will do the required tillage on the land and then carry out mechanized transplanting to avoid the imprecise transplanting row spacing caused by wheel slippage as much as possible before the land preparation. If wheel slippage still occurs in transplanting operation after land preparation, our preset index transplanting plant spacing is in the range of 100 mm (1±10%), which is set in advance to avoid the error caused by changing the problem and resulting in imprecise transplanting. For this problem, the project team subsequently developed an electrically driven control system to further solve the error caused by wheel slip or other factors by adding sensors and developing a control system, although the technology is still under development. The method proposed and the device designed in this paper, the impact on the operational performance caused by wheel slip after PTO transmission is within the allowable error range of the operational evaluation index. We hope that the experts can give their opinions and suggestions to help us solve the problem better. Thank you.
Point 2: Experimental design: What about the other factors such as moving belt speed, and rotary tiller parameters? Why these factors were not taken into account.
Response 2: In the early stage of machine design, the speed of the conveyor belt and the forward speed of the machine have been initially matched by the transmission ratio, so the speed of the conveyor belt is not considered; the parameters of the rototiller are important conditions for the monopoly operation, and the transplanting is completed on the basis of the monopoly operation, and the research on rototilling and monopoly technology has been carried out in the early stage, which meets the requirements of transplanting technology, so the parameters of the rototiller are not considered in this paper. Thank you.
Point 3: How the optimum level combination was interpreted from Table 3?
Response 3: According to page 127 in the reference book on experimental design and data processing, orthogonal tests are conducted by statistically analyzing the test results of these few test protocols, and the resulting superior protocols are not necessarily included in these few test protocols, so further field validation tests are needed. Thank you.
Point 4: The function of screw covering mechanism needs to be explained?
Response 4: Screw covering mechanism is the key component of sweet potato transplanter, with the advantages of stable and adjustable mulching volume, its function and design criteria are the important design basis in this paper, so it needs to be briefly introduced. Thank you.
Point 5: Comparison of the developed planter with any conventionally available potato seedling transplanting machine is missing.
Response 5: Comparison with the sweet potato transplanter commonly used at this stage has been added in the discussion. Thank you.
Point 6: Table 1: These (Row numbers 3, 4, 5, 6, and 8) are basically the performance parameters of the machine and not the technical specifications. Please delete it from this section and provide the proper detailed technical specifications of your developed machine.
Response 6: Performance parameters have been removed and the relevant technical specifications of the developed machine have been provided. Thank you.
Point 7: Nothing has been discussed about the rotary tiller mounted at the most front. What type of blades were used, rotor speed, depth of tillage etc.
Response 7: The key technology research of rotary tillage and monopoly has been carried out in the previous period, and the rotary tillage and monopoly technology has met the requirements of sweet potato transplanting. In this paper, the key technology research of sweet potato transplanting has been carried out, so the rotary tiller has not been discussed. Thank you.
Point 8: What is GB/T 5262? Is it a test code? Please give proper reference.
Response 8: GB/T 5262 is measuring methods for agricultural machinery testing conditions-General rules. It has been added to the references. Thank you.
Point 9: There should be a space between the numerical value and the unit. For example it should be 20-30 cm not 20-30 cm. Please correct it for the entire manuscript.
Response 9: The entire manuscript has been completely corrected. Thank you.
Point 10: Lines 132-153 should be moved to section 2.1.
Response 10: Lines 132-153 have been moved to Section 2.1. Thank you.
Point 11: Lines 157-158: What do you mean by length of planting and how it is a design criteria for deciding the width of conveyor belt? Please discuss here.
Response 11: Sorry, I didn't express this clearly. The length of planting is the total length of potato seedlings. If the width of the conveyor belt is smaller than the length of the seedlings, it will lead to unstable seedling transport and affect the planting effect, so the width of the conveyor belt is designed to be 250 mm according to the length of the transplanted seedlings. Thank you.
Point 12: Grooves should be shown in the Figure 6 for better understanding.
Response 12: Thank you for your valuable advice, Figure 6 has shown the groove.
Point 13: The headings of sections 3.4.1 and 3.4.2 are same. Please rectify.
Response 13: Thank you for your valuable advice, based on your comments, this issue has been corrected.
Point 14: Lines 305-315: Write the entire methodology in the correct format and tense. It seems like some instructions have been given to perform the tests.
Response 14: Thank you for your valuable advice, writing specifications have been corrected.

Reviewer 2 Report
When reading the introduction, one gets the impression that the issues raised in the publication are very local. And it is not so. While the biological issues concern China in particular, in terms of technology and machinery, the presented solutions may have a global reach. Therefore, the study of the bibliography should be significantly enriched.
Detailed comments:
(1) Avoid duplicating keywords that appear in the title.
(2) The kinematic diagrams in Figure 3 should be in accordance with the international standard ISO 3952.
(3) Instead of describing the dimensions - lines 146-149 - it is better to dimension Figure 5. A similar remark applies to Figure 7.
(4) Units of physical quantities throughout the article should be written in exponential form.
(5) The rotational speed should be given in units according to the international SI system.
(6) The photos in Figure 9 should be labeled and described.
Author Response
Point 1: Avoid duplicating keywords that appear in the title.
Response 1: Thank you for your valuable advice, this issue has been modified.
Point 2: The kinematic diagrams in Figure 3 should be in accordance with the international standard ISO 3952.
Response 2: The kinematic diagrams in Figure 3 has been modified according to the international standard ISO 3952. Thank you.
Point 3: Instead of describing the dimensions - lines 146-149 - it is better to dimension Figure 5. A similar remark applies to Figure 7.
Response 3: Thank you for your valuable advice. However, if the dimensions are marked in the drawing, the part structure cannot be visualized.
Point 4: Units of physical quantities throughout the article should be written in exponential form.
Response 4: The units of physical quantities throughout the article has been written in exponential form. Thank you.
Point 5: The rotational speed should be given in units according to the international SI system.
Response 5: The speed unit has been changed to rpm. Thank you.
Point 6: The photos in Figure 9 should be labeled and described.
Response 6: The photos in Figure 9 has been labeled and described. Thank you.

Round 2
Reviewer 1 Report
- The authors may add the ‘Response 1’ in their manuscript to indicate the problem of wheel slip they encountered in the field along with the solution (i.e. electrically driven control system) to help future researchers in this regard.
I want to suggest following paper which could be of help for future research.
Nataraj, E., Sarkar, P., Raheman, H., & Upadhyay, G. (2021). Embedded digital display and warning system of velocity ratio and wheel slip for tractor operated active tillage implements. Journal of Terramechanics, 97, 35-43.
- Table 2: Operating speed is mentioned as 100 m/s. Please correct this data.
Author Response
Point 1: The authors may add the ‘Response 1’ in their manuscript to indicate the problem of wheel slip they encountered in the field along with the solution (i.e. electrically driven control system) to help future researchers in this regard.
I want to suggest following paper which could be of help for future research.
Nataraj, E., Sarkar, P., Raheman, H., & Upadhyay, G. (2021). Embedded digital display and warning system of velocity ratio and wheel slip for tractor operated active tillage implements. Journal of Terramechanics, 97, 35-43.
Response 1: Thank you for your valuable comments. The wheel slip problem has been added in the discussion. Solutions and references for future research are provided. Thank you.
Point 2: Table 2: Operating speed is mentioned as 100 m/s. Please correct this data.
Response 2: Sorry, the operating speed is incorrectly stated, it has been corrected to ≥0.1 m/s. Thank you.
